

# Microbiome analysis of Pacific white shrimp gut and rearing water from Malaysia and Vietnam: implications for aquaculture research and management

Muhammad Zarul Hanifah Md Zoqratt[1,2,*], Wilhelm Wei Han Eng[1,2], Binh Thanh Thai[3], Christopher M. Austin[1,2,4,5] and Han Ming Gan[1,2,4,5,*]

[1] School of Science, Monash University Malaysia, Petaling Jaya, Selangor, Malaysia
[2] Genomics Facility, Tropical Medicine and Biology Platform, Monash University Malaysia, Petaling Jaya, Selangor, Malaysia
[3] Fisheries and Technical, Economical College, Dinh Bang, Tu Son, Vietnam
[4] Centre for Integrative Ecology, School of Life and Environmental Sciences, Deakin University, Geelong, Victoria, Australia
[5] Deakin Genomics Centre, Deakin University, Geelong, Victoria, Australia
[*] These authors contributed equally to this work.

Corresponding author
Han Ming Gan,
han.gan@deakin.edu.au

## ABSTRACT

Aquaculture production of the Pacific white shrimp is the largest in the world for crustacean species. Crucial to the sustainable global production of this important seafood species is a fundamental understanding of the shrimp gut microbiota and its relationship to the microbial ecology of shrimp pond. This is especially true, given the recently recognized role of beneficial microbes in promoting shrimp nutrient intake and in conferring resistance against pathogens. Unfortunately, aquaculture-related microbiome studies are scarce in Southeast Asia countries despite the severe impact of early mortality syndrome outbreaks on shrimp production in the region. In this study, we employed the 16S rRNA amplicon (V3–V4 region) sequencing and amplicon sequence variants (ASV) method to investigate the microbial diversity of shrimp guts and pond water samples collected from aquaculture farms located in Malaysia and Vietnam. Substantial differences in the pond microbiota were observed between countries with the presence and absence of several taxa extending to the family level. Microbial diversity of the shrimp gut was found to be generally lower than that of the pond environments with a few ubiquitous genera representing a majority of the shrimp gut microbial diversity such as *Vibrio* and *Photobacterium*, indicating host-specific selection of microbial species. Given the high sequence conservation of the 16S rRNA gene, we assessed its veracity at distinguishing *Vibrio* species based on nucleotide alignment against type strain reference sequences and demonstrated the utility of ASV approach in uncovering a wider diversity of *Vibrio* species compared to the conventional OTU clustering approach.

## INTRODUCTION

*Litopenaeus vannamei* (Boone, 1931), also known as the Pacific white shrimp or Whiteleg shrimp, is a major aquaculture commodity with a production of 3.69 million tonnes valued at 18 billion USD revenue (*FAO, 2016*). In recent years, significant outbreaks of acute hepatopancreatic necrosis disease (AHPND), also known as early mortality syndrome (EMS) have been reported in a number of white shrimp-producing countries. EMS was first reported in China in 2009 and subsequently spread to Southeast Asian countries including Vietnam, Malaysia, and Thailand (*Foo et al., 2017*; *Kondo et al., 2014*; *Tran et al., 2013*). The causative agent of EMS has been reported to be *Vibrio parahaemolyticus* strains harbouring a plasmid containing the *pirA-* and *pirB-* like genes encoding for toxins capable of severely damaging the shrimp gut (*Han et al., 2015*; *Lee et al., 2015*). To date, EMS has caused an estimated one billion USD of losses to the shrimp industry worldwide (*De Schryver, Defoirdt & Sorgeloos, 2014*; *Lee et al., 2015*).

Monitoring and control of pond water quality play a crucial role in managing and preventing disease outbreak in aquaculture. However, the current practice of water quality monitoring usually focuses on the measurement of chemical and physical parameters such as oxygen, pH, temperature, salinity, turbidity and nitrogen compounds. The importance of microbial communities in influencing or responding to variation in aquaculture pond water quality has only been recognized in recent years (*Bentzon-Tilia, Sonnenschein & Gram, 2016*). This is especially relevant to managing the water quality of aquaculture ponds and their cultured biomass because microbes carry out important biological services in aquaculture environment including nutrient cycling, probiotic/pathogenic activity and nutrient acquisition in addition to potentially acting as a rapid biological indicator of critical chemical changes in the rearing water (*Cardona et al., 2016*; *Cornejo-Granados et al., 2017*; *Costa, Pérez & Kreft, 2006*; *Emerenciano, Gaxiola & Cuzon, 2013*; *Grotkjær et al., 2016*; *Jinbo et al., 2017*; *Liu et al., 2015*; *Wright, Konwar & Hallam, 2012*; *Zeng et al., 2017*; *Zhu et al., 2016*). Microbes can also be used to improve the water quality of ponds. For example, adding denitrifying bacteria to biofilters has been shown to reduce the concentration of ammonia and its immediate derivatives, which are detrimental to shrimp health (*Saffran et al., 2001*).

Recognizing the importance of microbial biomass and diversity on the health and production of cultured invertebrates, several microbiome studies have analysed the gut microbiome of wild-caught shrimps (*Cornejo-Granados et al., 2017*; *Phayungsak et al., 2018*; *Rungrassamee et al., 2016*) as well as cultured shrimps under different abiotic and biotic factors. However, most studies have been restricted to a specific country especially China, focusing only on one or very few localized ponds (*Cornejo-Granados et al., 2017*; *Huang et al., 2016*; *Rungrassamee et al., 2016*; *Tang et al., 2014*; *Xiong et al., 2015*; *Zhang et al., 2014*; *Zhu et al., 2016*). A study in Thailand used denaturing gradient gel electrophoresis profiling and barcoded pyrosequencing to demonstrate a greater survivability of *Litopenaeus vannamei* and improved the resilience of its gut microbiome upon *Vibrio harveyi* exposure relative to that of *Penaeus monodon* (*Rungrassamee et al., 2016*). Another more recent study in Vietnam utilised a standard Illumina 16S rRNA gene amplicon sequencing method to

investigate the effect of EMS outbreak on the microbial interaction networks in shrimp guts (*Chen et al., 2017b*). Thus, despite the emergence of Southeast Asia (SEA) as an aquaculture hub, studies on any significant geographic scale are relatively scarce in this region. Further, to our knowledge, all recent shrimp aquaculture microbiome studies still employ the operational taxonomic units (OTU) clustering approach in marker-gene data analysis despite recent calls for the replacement of this approach with exact sequence variants which can resolve single-base differences among biological sequences in the sample thus providing a more comprehensive and accurate view of microbial communities (*Callahan, McMurdie & Holmes, 2017*; *Eren et al., 2014*; *Utter, Mark Welch & Borisy, 2016*).

To initiate a more broad-based investigation of shrimp microbiomes directly relevant to the aquaculture industry in SEA, we performed Illumina 16S rRNA gene amplicon sequencing of *L. vannamei* guts and rearing water from aquaculture farms located in two SEA countries with contrasting climates at the time of sampling i.e., Malaysia (warm and humid, 30 °C) and Vietnam (cool and dry, 20 °C). For the first time in a shrimp aquaculture microbiome study, we applied the recently advocated ASV-method (*Edgar, 2016*) to: (1) compare shrimp intestinal microbial diversity and their pond environments; (2) compare these microbial communities between Malaysia and Vietnam; and (3) assess the performance of the 16S rRNA V3–V4 hypervariable region and clustering approaches in capturing the genetic diversity of *Vibrio*.

## METHODS

### Sample collection

Sampling in Vietnam was performed at two separate shrimp farms in the Quang Ninh province (approximately 20 km apart), while sampling in Malaysia was performed at a large shrimp farm with multiple pond systems located in Perak state (Table 1). Two mL of pond water was sampled from 2–4 distant location (corners) of each pond, pelleted via centrifugation at 7,000 rpm for 10 min and resuspended in RNA/DNA shield (ZymoResearch, Irvine, CA, USA). Shrimp intestinal samples were collected by dissecting out the shrimp guts followed by homogenization in RNA/DNA shield (ZymoResearch, Irvine, CA, USA). Sampling was performed with the permission and under the supervision of the aquaculture manager from the respective farms. No field permit was required for this study because samples were collected from private fields. As per the request of the aquaculture manager, exact sampling location of some farms was not disclosed in this study to protect the identity of the farm.

### DNA extraction, amplification, purification and sequencing

Genomic DNA was extracted from the RNA/DNA shield lysate using DNA Clean & Concentrator$^{TM}$-5 (ZymoResearch, Irvine, CA, USA) according to the manufacturer's instructions. The V3–V4 region of the 16S rRNA gene was amplified using forward primer 5′–TCGTCGGCAGCGTCAGATGTGTATAAGAGACAGCCTACGGGNGGCWGCAG –3′ and reverse primer 5′–GTCTCGTGGGCTCGGAGATGTGTATAAGAGACAG GACTACHVGGGTATCTAATCC–3′ containing partial Illumina Nextera adapter. PCR reaction (∼10 ng input DNA/ reaction) and barcode incorporation were performed as

**Table 1  Summary of field collection and sampling design.**

| Farm | # Pond (Replicate per pond) | # Shrimp Sampled | Location | Collection date | Mean temperature (°C) | Reported age (days post hatching) |
|---|---|---|---|---|---|---|
| Aquaculture Research Station of Fisheries College | 4 (2–3) | 6 | Quang Yen, Quang Ninh, Vietnam | December 2015 | 20 | 97 |
| Private | 1 (4) | 4 | Quang Yen, Quang Ninh, Vietnam | December 2015 | 20 | 115 |
| Private | 9 (2–3) | 14 | Sitiawan, Perak, Malaysia | March 2016 | 30 | 65 |

previously described (*Watts et al., 2017*). Constructed libraries were quantified, normalized, pooled, denatured and subsequently sequenced on the Illumina MiSeq (Illumina, San Diego, CA, USA) located at Monash University Malaysia Genomics Facility using a $2 \times 250$ bp run configuration.

## Sequence data analysis and observation table construction

Primer sequences corresponding to the 16S rRNA gene were removed from the raw paired-end reads using Cutadapt (*Martin, 2011*). Trimmed forward and reverse reads were overlapped with fastq_mergepairs followed by quality and length filtering with fastq_filter (maximum expected error = 0.5; min length = 250 bp) as implemented in USearch10 (*Edgar, 2010*). Sequence dereplication and denoising was done using uNoise3 to generate amplicon sequence variants (ASVs) (*Edgar, 2016*). Aside from "-minuniquesize 2″ parameter during sequence dereplication, the processes of the pipeline were done using default parameters. Taxonomic assignment and observation table construction rarefied at 8,000 reads were performed in RDP classifier 2.12 and QIIME 1.9.1 (*Caporaso et al., 2010*; *Cole et al., 2009*). ASVs that failed RDP taxonomic assignment were re-classified using SINA 1.3.1 against SILVA SSU Ref database (release 132) with default parameters (Data S3 and Data S4) (*Pruesse, Peplies & Glöckner, 2012*; *Pruesse et al., 2007*). ASVs with lower than 0.01% fraction of the total normalized observation and/or identified as chloroplast were not included in subsequent analyses. Normalization of sequencing depth per sample (8,000 reads/sample), rarefaction curves construction (10 replicates/depth) as well as alpha diversity estimation (Simpson's evenness and Shannon diversity indices), were performed using the "core_diversity.py" python script in QIIME 1.9.1. Core genera of the shrimp gut microbiome were investigated and defined as genera with at least 0.1% relative abundance per sample in more than 50% shrimp samples. The prevalence and relative abundance of the shrimp gut core genera were visualised using the R *ggplot2* package (*Wickham & Wickham, 2007*).

## Principal coordinates analysis and relative differential abundance analysis

The proportion of sequences assigned to the lowest possible taxonomic level was calculated based on the rarefied observation table and the implemented taxonomic assignment method mentioned above. Principal coordinates analysis (PCoA) was also constructed based on weighted Unifrac and unweighted Unifrac using ordination method in R *phyloseq*

package (*Lozupone & Knight, 2005*; *McMurdie & Holmes, 2013*). The resulting PCoA plots were then visualized using R *ggplot2* package (*Wickham & Wickham, 2007*). Strength and significance of grouping were calculated using the compare_categories.py python script in QIIME which implements ANOSIM analysis using the default 999 permutations. Relative differential abundance test was also conducted at phylum and family levels using Tukey-Kramer post-hoc in conjunction with analysis of variance (ANOVA) statistical test in STAMP (*Parks et al., 2014*). Multiple hypothesis testing was done for the four generic groups namely Malaysian Farm, Malaysian Shrimp, Vietnamese Farm and Vietnamese Shrimp. A significant differential abundance of phylum distribution was defined as Benjamini–Hochberg-corrected probability $p$-value of $\leq 0.01$ and was observed only for the top 10 most abundant phyla. A significant differential abundance of family distribution was defined as Benjamini–Hochberg corrected $p$-value of $\leq 0.01$ and eta squared $\geq 0.3$.

## Comparison of *Vibrio* diversity using different clustering methods

*Vibrio* diversity was compared using different methods of 16S marker-gene data analysis, namely conventional operational taxonomic unit (OTU) clustering and amplicon sequence variants (ASV), using UParse and uNoise3 respectively (*Edgar, 2013*; *Edgar, 2016*). Except for -minuniquesize of 2 during sequence dereplication, both pipelines were conducted using default parameters. ASVs and OTUs assigned to the genus *Vibrio* with at least cumulative read abundance of more than 200 were retained for blastN similarity search ($E$-value $< 1e-100$) against 16S rRNA sequences of *Vibrio* type strain curated in EzBioCloud (as of 18th May 2018) (*Yoon et al., 2017*).

# RESULTS

## Shrimp intestinal and pond microbial communities are distinct

A total of 2,731,818 successfully merged reads were generated in this study with 2,144,192 reads (median of 32,648 reads/sample; min $= 9,948$; max $= 66,590$) confidently mapped to the ASVs. 92.12% and 7.77% of the mapped reads correspond to RDP-classified and SINA-classified ASVs respectively, while the remaining mapped reads belong to ASVs without confident taxonomic assignment at the kingdom rank. A majority of reads recovered from shrimp intestine and ponds were assigned to members from the phyla Proteobacteria, Actinobacteria, Bacteroidetes and Fusobacteria (Fig. 1). Significant differences in the relative abundance of bacteria phyla were observed among samples isolated from ponds and shrimp guts. Reads mapping to Actinobacteria and Bacteroidetes are more abundant in ponds ($p < 0.01$ and $p < 0.001$, respectively), while shrimp guts have a significantly higher relative abundance of Proteobacteria ($p < 0.001$). At a finer level, Malaysian shrimp intestinal microbiome contains more reads mapping to the phylum Fusobacteria ($p < 0.01$). Notable, this phylum is also near absent in three out of four Malaysian shrimps noted to be unhealthy based on morphological observation by the aquaculture manager (Fig. 1).

Rarefaction curves based on alpha diversity metrics, number of observed ASVs and PD (Phylogenetic diversity) whole tree, indicated that 8,000 sequences per sample are sufficient for capturing the alpha diversity of microbial communities in both shrimp guts and ponds. Inverse Simpson's and Shannon's indices showed higher species richness and

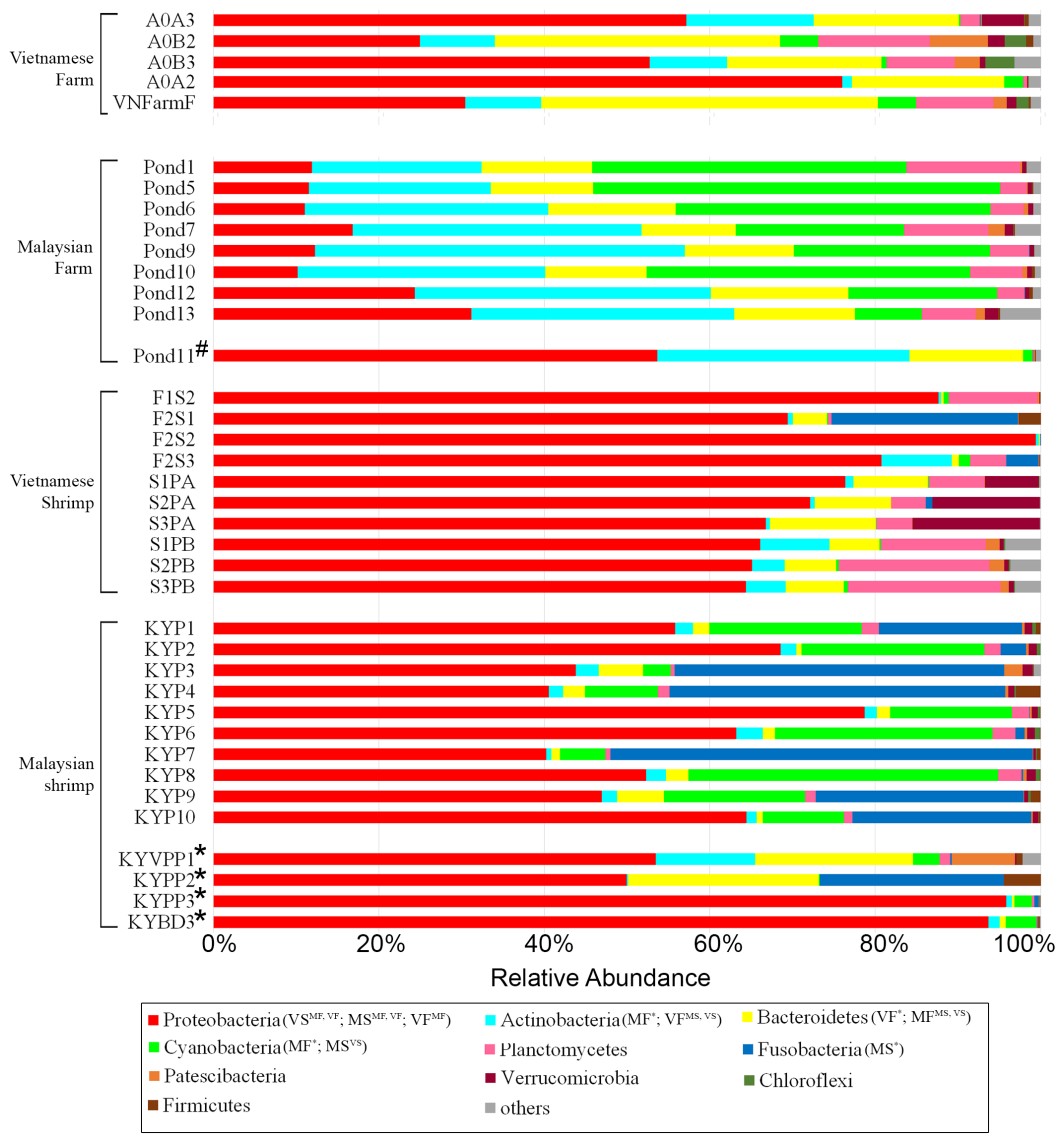

**Figure 1** **Distribution of abundant phyla in all samples.** Significantly abundant phyla are annotated at the legends. Phyla abundance of pond water replicates were collapsed according to the location of sampling (see Table S1). Sample type (MF, Malaysian rearing water; VF, Vietnamese rearing water; MS, Malaysian shrimp; VS, Vietnamese shrimp) with significantly higher phylum abundance than that of another group (Superscript) were shown in bracket next to their associated phylum legend. For example, Actinobacteria (MF*; VF$^{VS}$, MS) indicates that this phylum is significantly more abundant in Malaysian rearing water samples compared to all three other groups and that it is more abundant in Vietnamese rearing water samples compared to shrimp samples from both countries. Hash and asterisk signs next to $y$-axis labels indicate mussel-infested and suspected diseased samples, respectively.

evenness in the pond microbiome compared to that of shrimp gut (Figs. 2A and 2B). Beta diversity analyses based on both weighted and unweighted UniFrac indicated that shrimp gut and pond microbial communities from the same sampling site are significantly different (Vietnam: $R > 0.67$, $p$-value $< 0.001$; Malaysia: $R > 0.89$, $p$-value $< 0.001$) (Table S2). An
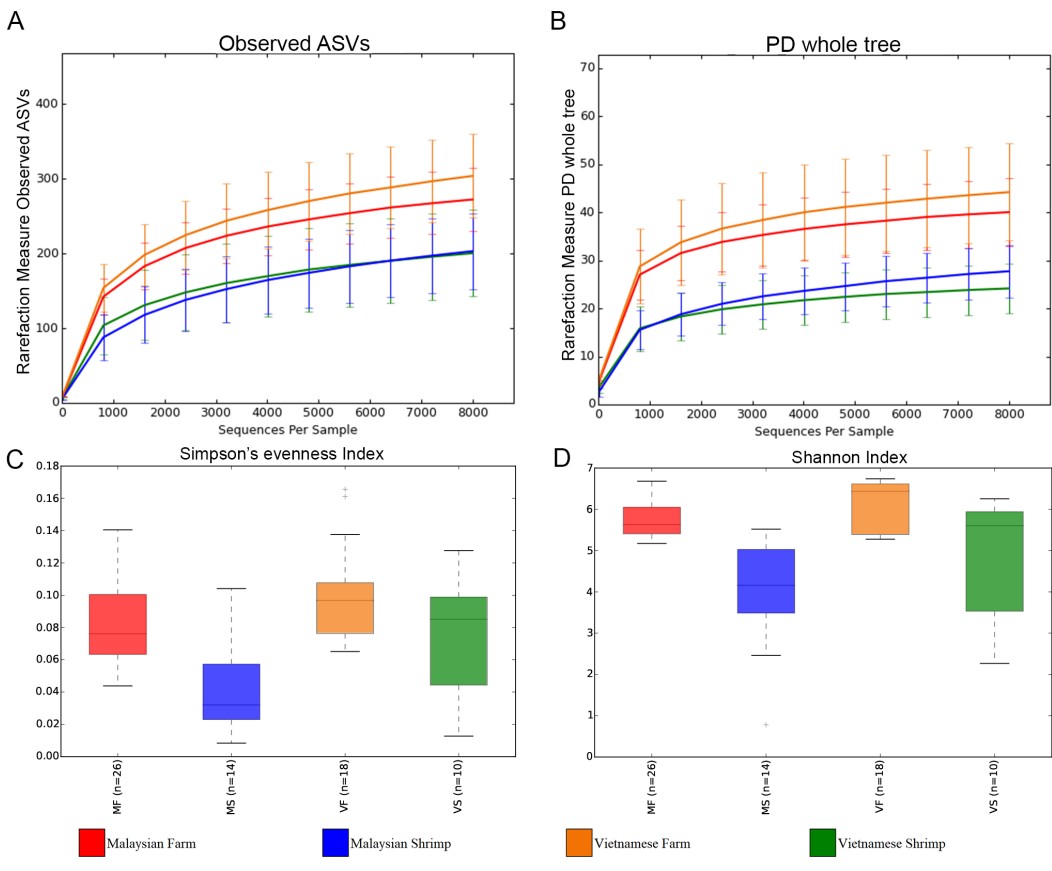

**Figure 2** **Rarefaction curves and alpha diversity plots of each sample group.** (A) Rarefaction curve constructed based on observed ASVs. (B) Rarefaction curve constructed based on phylogenetic distance (PD_whole_tree). (C) Shannon's evenness index box plot. (D) Shannon index box plot.

even stronger separation was also observed among pond samples from different sampling sites/ countries (Fig. 3, Table S2). Furthermore, samples from one of the Malaysian ponds (MF_Pond11) noted by the shrimp farmer to be infested by mussels (Table S1) was distinct from other Malaysian pond sample samples (Fig. 3). It is worth noting that rearing water samples collected from different parts of the same pond have minimal spatial variation in microbial composition as evidenced by the general tight clustering of pond replicates, suggesting homogenous microbial community in the rearing water and indicating that the sampling protocols are efficient for capturing pond diversity. Although the separation between Malaysian and Vietnamese shrimp gut samples was less obvious in both PCoA plots with occasional overlap, their microbial community structure appears to differ moderately ($R < 0.67$) with good statistical support ($p$-value $< 0.001$) based on ANOSIM analysis (Table S2).
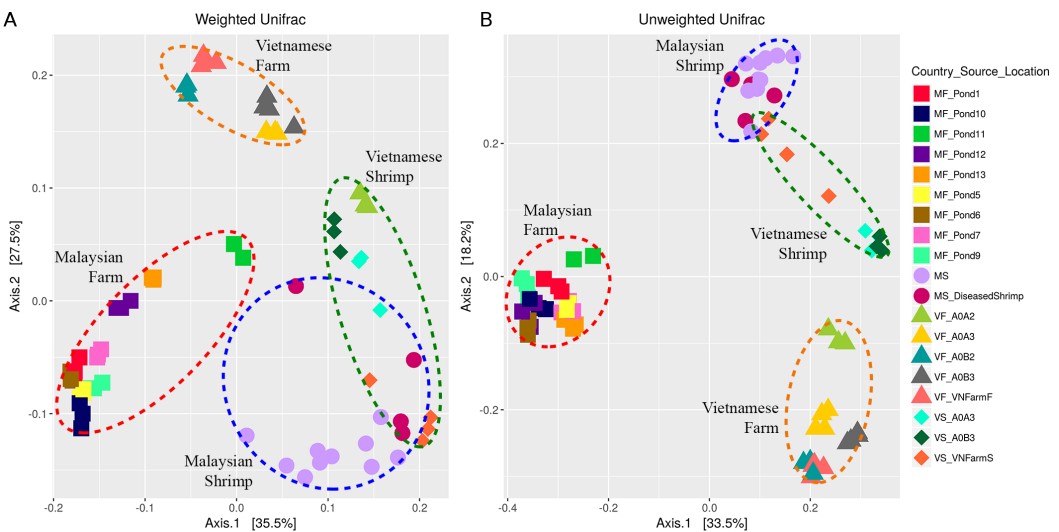

**Figure 3** **Principal component analysis of (A) weighted Unifrac and (B) unweighted Unifrac distances.**
Red circular outline shows 'Malaysian pond' cluster, yellow circular outline shows 'Vietnamese pond' cluster, blue circular outline marks 'Malaysian shrimp' cluster, and green circular outline represents 'Vietnamese shrimp sample. The sample points were coloured according to "Country_Source_Location" information (Table S1). Country and source information of each points were abbreviated; for example, Malaysian farm sample from Pond 1 was labelled as MF_Pond1.

## Comparison of relative abundance at the microbial family level reveals fine-level microbiota dynamics

Given that more than 70% and 85% of the reads derived from pond and shrimp intestinal samples, respectively, could be assigned to the family level (Data S3 and S4), we defined the core and unique microbiomes among sample groups at this taxonomic level and used STAMP to identify statistically significant differences in the relative abundance of microbial families. A total of six microbial families (Alcaligenaceae, Flavobacteriaceae, Microbacteriaceae, Acidimicrobiaceae and Rhodobacteraceae) are shared across shrimp and rearing water samples (Fig. 4A and Data S5) with seven and 10 microbial families uniquely present in Malaysian and Vietnamese rearing water, respectively, corroborating with their higher alpha diversity (Figs. 2C and 2D). In addition, four microbial families are uniquely shared by the Malaysian and Vietnamese rearing water samples, suggesting their common affiliation with shrimp rearing water. Certain microbial families are significantly more abundant in samples of the same isolation source, regardless of the country of origin. For example, Microbacteriaceae and Flavobacteriaceae are more abundant in pond water, while Vibrionaceae is highly enriched in the shrimp gut (Fig. 4B). On the other hand, Rhodobacteraceae is more abundant in Vietnamese samples (shrimp gut and rearing water) while Cyanobacteria-Family II is more abundant in Malaysian samples, suggesting possible affiliation to regional climate and/or pond environment. In addition, four microbial families (Acidomicrobiaceae, Actinomarinaceae, Comamonadaceae, and Fusobacteriaceae) showed significantly higher abundance only in a specific country and isolation source (Fig. 4B).

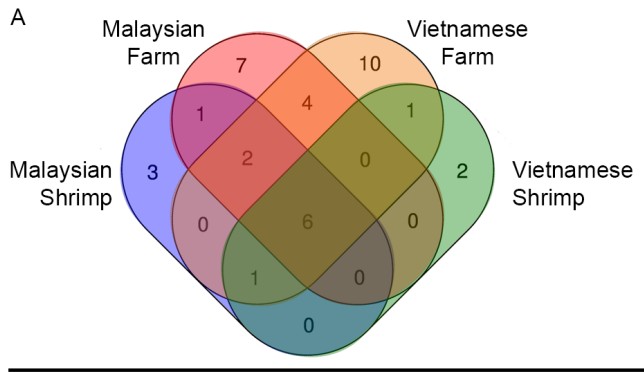

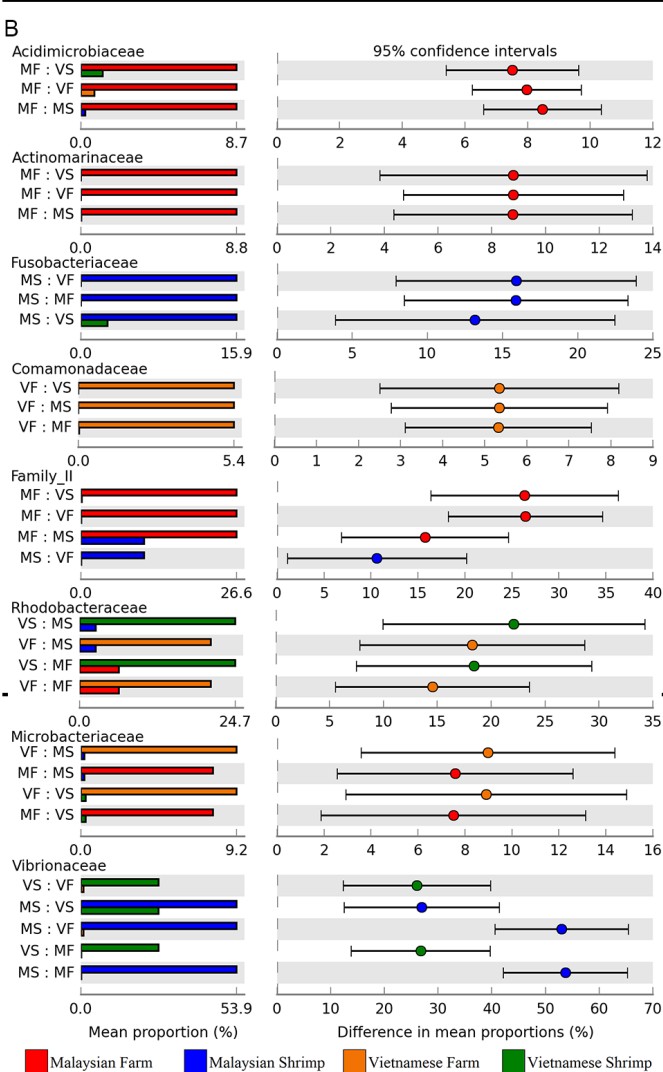

**Figure 4 Microbial community dynamics at the family level.** (A) Venn diagram illustrating the number of unique and overlapping microbial families among shrimp guts and rearing water. To be considered as present, a microbial family must be detected in at least 90% of the samples from the same group. (B) Tukey post-hoc pairwise comparison in conjunction with analysis of variance (ANOVA) between "Country_Source" groups of nine significant microbial families. Mean proportion values of families that are significantly different are shown in the bar plots on the left section, while the differences in the pairwise comparison mean proportion with 95% confidence interval are shown on the right section.

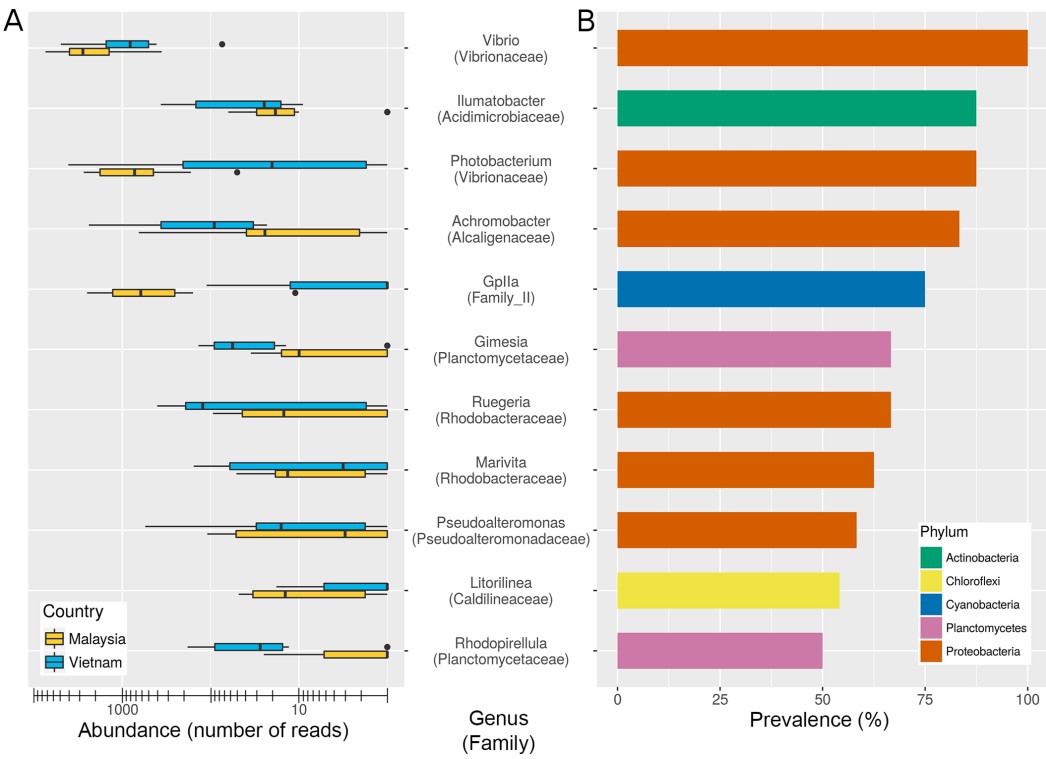

**Figure 5** **Abundance and prevalence of core genera in shrimp guts.** (A) Normalized read counts of the selected genera among Malaysian and Vietnamese shrimp gut samples (B) Percentage of shrimp gut samples harbouring the core genera

## High prevalence and relative abundance of ASVs belonging to the genera *Vibrio* in shrimp gut

16S rRNA reads corresponding to 11 bacterial genera belonging to five phyla (Actinobacteria, Chloroflexi, Cyanobacteria, Planctomycetes and Proteobacteria) were detected in more than 50% of the shrimp gut samples (Fig. 5) with *Vibrio* being the only genus present in all shrimp gut samples with relatively similar relative abundance across samples(mean/median relative abundance per sample = 33.7%/ 27.7%). On the contrary, *Photobacterium,* a genus related to *Vibrio* at the family level (Vibrionaceae) was detected in all but two Vietnamese shrimps (mean/median relative abundance per sample = 11.6%/ 6.5%). The prevalence of the core genera was independently correlated with relative abundance e.g., the higher the prevalence of a core genus, the more likely it is to have a higher relative abundance. *Rhodopirellua* and *Gimesia* all belonging to the phylum Planctomycetes are more prevalent and abundant in Vietnamese shrimps than in the Malaysian shrimps, consistent with statistical analysis showing a significantly higher abundance of Planctomycetes in Vietnamese shrimps compared to Malaysian shrimps (Fig. 1).

## Substantial underestimation of *Vibrio* diversity using the OTU clustering approach

The high cumulative abundance of reads assigned to the genus *Vibrio* in shrimp guts indicates that some members of this genus are endogenous to the shrimp gut microbiota. UPARSE using the default 97% sequence similarity cut-off setting identified two abundant OTUs assigned to *Vibrio*. This contrasts greatly with the ASV approach which identified substantially more biological sequences classified as *Vibrio* (Fig. 6, Data S3 and S4). A majority of the constructed ASVs do not have an exact sequence match to the constructed OTUs and more importantly, some could be assigned to a single *Vibrio* species (ASV22, *Vibrio jasicida* TCFB 0772[T]; ASV19, *Vibrio neocaledonicus* NC470[T]). OTU2 and OTU10 are identical in both sequence length and identity to ASV2 and ASV14, respectively. Similarity search of OTU2/ASV2 revealed an exact sequence match to *V. rotiferianus* LMG21460[T] and *V. campbellii* CAIM519[T], indicating a limitation to the use of V3–V4 hypervariable region in delimiting some *Vibrio* species (Fig. 6). The high ratios of ASVs-to-OTUs observed for OTU2 and OTU10 strongly suggests that imposing a fixed dissimilarity threshold using conventional OTU clustering underestimates the true microbial diversity for *Vibrio* species in this study and that resolving amplicon sequence variants (ASV) from amplicons data which is sensitive down to single-nucleotide differences, through read de-replication and error correction, substantially increased the number of observed *Vibrio* species from the identical dataset. Unlike the ASVs associated with OTU2, nearly all ASVs associated with OTU10 have at least two mismatches to known *Vibrio* species, indicating the presence of unculturable or yet-to-be-cultured *Vibrio* strains in the shrimp gut. Of even more interest, the ASV approach revealed the presence of *V. parahaemolyticus* (ASV235 in Fig. 6) that was missed by the OTU clustering approach presumably due to its overall low relative abundance across samples (Table S5).

## DISCUSSION

Despite the immense scale of shrimp aquaculture in South East Asia and the major impacts of aquaculture disease outbreaks in tropical regions, we are only starting to understand the microbial composition of the shrimp guts and their relationship to rearing water in the region (*Leung & Bates, 2013*). Most shrimp-related microbiome studies have been limited to a few farms in a particular country; most of which have been conducted in countries outside of SEA such as China and Mexico. *Litopenaeus vannamei* microbiome studies have so far investigated the microbial composition of wild-type shrimps serving as an important baseline for future comparative studies (*Cornejo-Granados et al., 2017*) as well as the impacts of disease exposure (*Chen et al., 2017b*; *Cornejo-Granados et al., 2017*; *Jinbo et al., 2017*; *Rungrassamee et al., 2016*; *Xiong et al., 2015*; *Zhu et al., 2016*), developmental stages (*Huang et al., 2016*), nutrition (*Zhang et al., 2014*) and temperature (*Tang et al., 2014*) on shrimp intestinal microbiome. We have contributed new findings to the growing literature by providing the first data on gut and pond water microbiome of Malaysian cultured shrimps. Furthermore, we compared bacterial communities of shrimp guts and pond water from multiple aquaculture farms in two distinct climatic regions (Malaysia

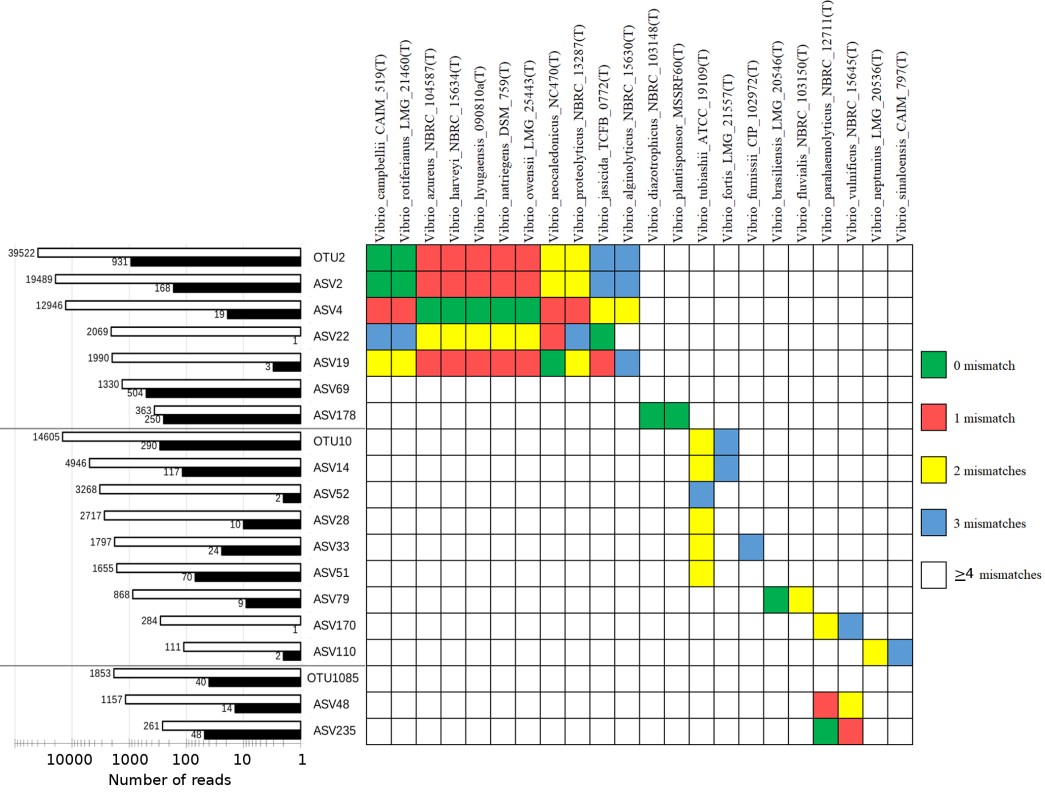

**Figure 6** **Cumulative normalized read counts of OTUs/ASVs classified as *Vibrio* and similarity table against selected type-strain *Vibrio* sequences.** Empty and filled bars indicate shrimp gut and rearing water samples, respectively. Different *Vibrio* groups consisting of one OTU and its associated ASVs were separated by grey horizontal lines.

and north Vietnam) while standardizing DNA extraction and sequencing protocols. This provides a new perspective on our current understanding on the range of normal microbial composition of a healthy shrimp gut microbiome community. Our principal findings provide evidence supporting microbiota plasticity in shrimp ponds, but in contradistinction, find a much more limited diversity of the adult shrimp intestinal microbiota.

Comparison of the pond water microbiome between the Malaysian and Vietnamese samples reveals significant dissimilarities at the phylum level as shown in Fig. 1. Although Planctomycetes was identified as a significant phylum in this study, it was not commonly found in other metagenomic studies of similar environments (*Li et al., 2016*; *Xiong et al., 2015*; *Zeng et al., 2017*) (Table S4). The only sample site with near-zero relative abundance of Plantomycetes was a mussel-infested Malaysian aquaculture pond (Pond11 in Fig. 1). Mussels are known ecosystem engineers that can substantially modify their habitats through biological processes such as sediment filtration and biodeposition of fecal matters (*Bril et al., 2014*). The presence of mussels has been reported to cause to a 4-fold decrease in the relative abundance of Plantomycetes in a freshwater system in Mississippi, USA (*Black, Chimenti & Just, 2017*).

Although Flavobacteriaceae and Microbacteriaceae are prevalent in the rearing water and shrimp gut samples, they exhibited significantly higher relative abundance in the rearing water samples. Albeit initially associated with a fairly general ecological function e.g., simple mineralization-based commensalism (*Kirchman, 2002*), emerging evidence suggests that some members within the marine Flavobactericeae clade are algal-associated species that exhibit growth promoting and inhibiting effects to its host and other algal species, respectively (*Bowman, 2006*). Genera such as *Cellulophaga, Psychroserpens* and *Formoasa* have been previously reported to produce toxic secondary metabolites against dinoflagellates, commonly associated with algal bloom (*Adachi et al., 2002*; *Egan et al., 2000*). Thus, the significant abundance of Flavobacteriacea in both rearing water could be linked to the natural occurrence of algal and diatom species in the rearing water some of which were co-amplified by the V3–V4 primers in this study (Data S3 and Table S3). On the contrary, most described members from the family Microbacteriaceae were not associated with marine environmental and were typically isolated from the terrestrial environment (*Evtushenko & Takeuchi, 2017*). High abundance of Microbacteriaceae (unclassified at the genus level) in shrimp rearing water particularly during the post-larvae stage has been previously observed in a commercial marine shrimp hatchery in Hainan, China (*Zeng et al., 2017*) which was suggested to be a temporal-specific bacterial family caused by changes in the shrimp diet during different growth stages. On the contrary, four microbial families namely, Cryomorphaceae, Rhodospirillaceae, Bacteriovoracaceae and Saprospiraceae, are exclusively found across both Malaysian and Vietnamese shrimp rearing water samples, indicating their specific adaptation to shrimp rearing water or more generally the marine aquatic environment. For example, members of the family Rhodospirillaceae are purple non-sulfur and mostly nitrogen-fixing photosynthetic bacteria. Their absence in the shrimp gut is consistent with their strict requirement for light to grow and proliferate which is not sufficiently present in the shrimp gut environment.

Despite the conspicuous difference in the shrimp pond microbiota between two countries, the shrimp intestinal microbiota are more similar to each other which is presumably due to host selection for microbial strains that adapt to or exploit the shrimp gut environment as corroborated by their lower alpha diversity indices compared to that of rearing water samples (*Xiong et al., 2017*). However, the presence of several microbial families in both shrimp and rearing water samples indicates that the shrimp gut microbiota maybe significantly affected by the microbial communities present in their aquatic environment e.g., rearing water and pond sediments (*Chen et al., 2017a*; *Cornejo-Granados et al., 2017*) as opposed to being maternally influenced as observed in some mammals and other animals with forms of parental care (*Jakobsson et al., 2014*; *Kohl & Dearing, 2012*; *Zhang et al., 2014*). Moulting e.g., shedding of shrimp ectodermal gut tissue also provides a new opportunity for shrimp stomach and guts to be colonised by the bacterial community of the pond (*Moss, LeaMaster & Sweeney, 2000*). Furthermore, crustaceans, including *L. vannamei*, also consume their exuvia, which provides another opportunity for microbial recolonization of the shrimp guts and also the transmission of intestinal microbiome across shrimps throughout their developmental stages (*Martínez-Córdova & Peña Messina, 2005*).

Although the micro-clustering of shrimp gut samples based on country and/or farm of origin may be associated with the difference in their respective growth environment, variation in developmental stage may also contribute to the observed clustering as the shrimps in this study were collected at different adult growth stages (Table 1). The abundance of members from the family Fusobacteriaceae is highly dependent on the shrimp developmental stage with a near-zero abundance in young shrimp larvae gut and subsequently making up a substantial portion of the microbiome in the adult stage (75-day post-hatching) (*Chen et al., 2017b*; *Zeng et al., 2017*). Intriguingly, although most of the Vietnamaese shrimps were 97-day post-hatching during sampling, the low abundance and prevalence of Fusobactericeae in the their guts indicates microbiome resemblance to that of younger shrimps (*Zeng et al., 2017*). In contrast, Fusobacteriaceae is prevalent in Malaysian shrimps that are relatively young e.g., 65-day post- hatching. However, it is worth noting that Sitiawan, the closest city to where the Malaysian farm is located, is warm (30 °C) throughout the year. Such a climate may support faster shrimp growth thus enabling them to reach adulthood earlier (*Kumlu, Türkmen & Kumlu, 2010*; *Wyban, Walsh & Godin, 1995*). Members from the family Fusobacteriaceae are microaerotolerant to obligate anaerobic Gram-negative rods bacteria that derive energy through the fermentation of a variety of carbohydrates, amino acids and peptides. Such a metabolic profile is consistent with the higher prevalence and abundance of Fusobacteriaceae in mature shrimp intestinal systems that typically exhibit a better digestive ability (*Parte et al., 2011*; *Schock et al., 2013*). Unfortunately, despite exhibiting a wide ecological diversity as evidenced by their diverse isolation source, the symbiotic relationship of Fusobacteriaceae towards its host has yet been properly demonstrated (*Nelson, Rogers & Brown, 2013*). Future work consisting of metatranscriptome and metagenome sequencing of the shrimp gut microbiota will be necessary to shed light on the role of Fusobactericeae in the shrimp gut.

*Vibrio* and *Photobacterium* belonging to the Vibrionaceae family are both abundant and prevalent in nearly all of the shrimp samples, an observation that is consistent with previous reports (*Cornejo-Granados et al., 2017*; *Rungrassamee et al., 2016*; *Xiong et al., 2017*). In contrast, *Zeng et al. (2017)* did not identify any *Vibrio*-specific OTUs in their sampling. Such anomalies are unlikely to be biological but rather due to technical and analytical differences, such as the choice of the 16S rRNA gene region sequenced (V4- vs. V3–V4-hypervariable region) and bioinformatic analysis settings. High abundance and prevalence of *Vibrio* and *Photobacterium* genera in shrimp guts suggest that they are more likely to be endogenous rather than pathogenic strains (*Kriem et al., 2015*). However, this also reflects the persistence and adaptation of members from these genera to the shrimp gut environments and may potentially explain the susceptibility of shrimps to non-native pathogenic *Vibrio* and *Photobacterium* strains (*Kondo et al., 2014*; *Wang & Chen, 2006*).

The use of ASVs reveals a wider diversity of *Vibrio* species, suggesting that previous shrimp microbiome analyses that employed the common 97% similarity cut-off for clustering will risk masking the true *Vibrio* diversity in the shrimp gut (*Callahan, McMurdie & Holmes, 2017*; *Chen et al., 2017b*). Fortuitously, despite the observed low resolution of the V3–V4 hypervariable region for *Vibrio* species, this region appears to be distinct in *V. parahaemolyticus,* which exhibits at least two diagnostic nucleotides that are absent from

all known type strains of *Vibrio* species (Fig. 6). The lack of an OTU with exact match to ASV235 suggests that analysis using the OTU clustering approach will fail to report the presence of *V. parahaemolyticus* and/or undescribed *Vibrio* strains sharing the same 16S rRNA gene sequence with *V. parahaemolyticus* if they are present at low abundance in the dataset. This can have critical implications for aquaculture microbial management especially in the early detection of *V. parahaemolyticus* infection. Since shrimp gut can harbour both pathogenic and native *Vibrio* species, complementing 16S rRNA-based amplicon sequencing with an alternative genetic marker such as *pyrH* may enable a more accurate quantification of *Vibrio* diversity and abundance in aquaculture environment (*Tall et al., 2013*; *Thompson et al., 2005*). Given the high diversity of shrimp gut-associated *Vibrio* as revealed for the first time by the ASV approach, shallow shotgun metagenome sequencing will also be instructive to obtain species/strain-level taxonomic resolution of the abundant endogenous microbes in shrimp guts particularly those belonging to the genera *Vibrio* and *Photobacterium.*

Shrimp gut microbiomes vary due to biological differences (shrimp strains), differences in environmental or farming practice (temperature, diet, probiotic, wild capture) or even biases from different laboratory procedures such as the sequencing platform and the different partial 16S sequence target (*Cornejo-Granados et al., 2017*; *Tremblay et al., 2015*). Considering the crucial functions undertaken by microbial communities and the potential use of the pond microbiome for pond health surveillance, investment in measuring a wide variety of chemical and physical parameters would allow us to better correlate the relationship between microbiomes and rearing water quality and therefore improving our understanding of aquaculture microbiomes.

## CONCLUSIONS

Using a standardized Illumina 16S rRNA amplicons sequencing protocol, we report for the first time, amplicon sequence variants (ASV)-based analysis of aquaculture rearing water and shrimp gut microbiota from two South East Asia countries with different climates. Despite substantial difference in the microbial composition of shrimp rearing water between farms in Malaysia and Vietnam, adult shrimp guts are more similar and exhibit a genus level core microbiome with the genus *Vibrio* being the most prevalent and abundant group. In addition, compared to OTU clustering approach, the ASV method improved the identification of closely related and/or rare *Vibrio* species, which is of relevance to the shrimp aquaculture industry. The high abundance of *Vibrio* in shrimp gut also suggests that some *Vibrio* species are endogenous and non-virulent to shrimps with functional and ecological roles that remain to be elucidated in the future.

## ACKNOWLEDGEMENTS

We thank the Monash University Malaysia Genomics Facility for the provision of computational resources. We are also extremely grateful to the aquaculture managers for providing access to their farms and sharing information.

### Funding

This work was supported by the Tropical and Medicine Biology Platform, Monash University. The funders had no role in study design, data collection and analysis, decision to publish, or preparation of the manuscript.

### Grant Disclosures

The following grant information was disclosed by the authors:
Tropical and Medicine Biology Platform, Monash University.

### Competing Interests

The authors declare there are no competing interests.

### Author Contributions

- Muhammad Zarul Hanifah Md Zoqratt performed the experiments, analyzed the data, prepared figures and/or tables, authored or reviewed drafts of the paper, approved the final draft.
- Wilhelm Wei Han Eng performed the experiments, approved the final draft.
- Binh Thanh Thai conceived and designed the experiments, contributed reagents/materials/analysis tools, approved the final draft.
- Christopher M. Austin conceived and designed the experiments, contributed reagents/materials/analysis tools, authored or reviewed drafts of the paper, approved the final draft.
- Han Ming Gan conceived and designed the experiments, performed the experiments, analyzed the data, prepared figures and/or tables, authored or reviewed drafts of the paper, approved the final draft.

### DNA Deposition

The following information was supplied regarding the deposition of DNA sequences:
All FastQ raw data may be accessed through SRA accession number SRP126985 or NCBI BioProject PRJNA422950.

### Data Availability

FASTA files for OTU and ASV in addition to their taxonomic assignments are available in the Supplemental File.

### Supplemental Information

Supplemental information for this article can be found online at http://dx.doi.org/10.7717/peerj.5826#supplemental-information.

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
