# Peer review of "Microbiome analysis of Pacific white shrimp gut and rearing water from Malaysia and Vietnam: implications for aquaculture research and management"

_PeerJ, doi:10.7717/peerj.5826_

## Round 0.1 · original submission · Minor Revisions

All reviewers agree that the revised manuscript has substantially improved. Nevertheless, I would like to ask you to address the remaining suggestions raised, including points related to methodology, presentation of data and interpretation.

Reviewer 1 ·

Basic reporting

The manuscript entitled “Microbiome analysis of Pacific white shrimp gut and rearing water from Malaysia and Vietnam: Implications for aquaculture research and management” aims at characterizing the microbiota from shrimp gut and pond water of farms located in Malasya and Vietnam using the amplicon sequence variants (ASV) method. The manuscript provides an interesting analysis of the whiteleg shrimp gut microbiota, and it has adequate experimental design and data analysis of sequences from V3-V4 16S rRNA region. Authors found a limited diversity of the adult shrimp gut microbiota as compared to the water from the same pond, which is presumably due to host selection for microbial strains. The details of the sequencing and annotation seemed appropriate. Importantly, the sequencing data are freely available for the scientific community. The statistical test applied to microbiota data is acceptable. Literature references are sufficient field background/context. The results are relevant results to hypotheses.

However, several major and minor points need to be addressed to improve the manuscript.

Experimental design

However, several major and minor points need to be addressed to improve the manuscript.

Major changes:

Lines 277-286:
It is probably due that Planctomycetes was not found in other studies due to the bioinformatics methodology (OTUs Vs ASVs). Please discuss in this regard.

Lines 350-355 and 385-387.
How many reads is the difference in the assignation to Vibrio species among using OTUs Vs ASVs?

The alpha diversity analysis showed a statistical difference between studied groups. However, the authors should specify how many replicates for the rarefactions curves were made to obtain the alpha diversity metrics.

Please clarify the cut-off of the filters used for quality filtering: which was the selected Phred quality score? The ambiguous bases were removed? A minimum length of the read was selected? Etc.

How many of the total sequenced reads (after quality filtering) matched against RDP database? How many reads were unclassified? Please discuss that. How many reads per sample were obtained?

Lines 236-241.
Please describe how many reads were assigned to Vibrio using OTUs and ASVs.

Authors claim that using ASV to detect Vibrio parahaemolyticus is an option for early detection of diseases such as EMS. However, the authors should demonstrate that utility using samples of microbiomes with really EMS diseased shrimps such as doi: 10.1038/s41598-017-09923-6 and 10.1038/s41598-017-11805-w.

Please include the age and weight (g) of shrimps collected in methods, and discusses the potential effects of the shrimp age in defining the difference of the microbiota among Malaysian and Vietnamese shrimps (Figure 3).

For a better comprehension of the shared and unique microbiomes between intestines and water from the same pond and between farms, the construction of Venn diagrams of shared ASVs at family and/or genus level will be very useful.

Validity of the findings

No comment.

Additional comments

Minor changes.

Lines 54-67:
There are several articles missed in the introduction about the microbiota from pond water and sediments of shrimp farms, i.e. doi: 10.1186/s12866-016-0770-z and others. I think that these works can be added to the cited background of the introduction section.

Lines 70-73:
There are at least two references of microbiota from wild-type shrimps, such as doi: 10.1371/journal.pone.0091853 and 10.1038/s41598-017-11805-w. These two references should be cited as wild-type shrimp microbiomes in these lines.

Line 103:
How was pelleted the sample, using centrifugation or gravity?

Lines 115-120:
How much was DNA concentration for PCR amplification used?

Line 266:
Please include the following two cites about the impact of diseases in L. vannamei microbiota in the sentence, doi: 10.1038/s41598-017-09923-6 and 10.1038/s41598-017-11805-w.

Lines 265-268:
There is at least one reference of L. vannamei microbiota from a wild-type environment (doi: 10.1038/s41598-017-11805-w). This reference should be cited as wild-type L. vannamei microbiome in these lines.

Lines 372-377.
I think that microbiota from pond sediments is also important with the species contribution to the gut shrimp bacterial diversity. Thus the authors could discuss that.

Figures 2-6.
A short description of what is shown in each panel and the symbols used are missing. For example, A, B, C and D symbols, and legends are missed in figure 2.

Figure 5.
The value of the X-axis is not clear. What is meaning 1000 or 10?

Figure 6.
In the left section, what the light and dark colors of the bars mean? What is the difference between these two colors?

Reviewer 2 ·

Basic reporting

The article is clear and well organized. The revisions have improved the manuscript.

Experimental design

This study uses appropriate analyses to address the question of how aquacultured shrimp microbiomes differ across geographic scales and from their water environment.

Validity of the findings

The authors have addressed my concerns regarding overstating some of their results and I believe that their findings are well supported.

Reviewer 3 ·

Basic reporting

Much improved writing from the first submission. No additional comments.

Experimental design

Experimental design is mostly the same but its description is much improved; details are better defined.
No additional comments

Validity of the findings

no comment

Additional comments

The authors have addressed my concerns from the prior submission. This is now a much better paper and will be more useful to the field.

I understand from the text why Supplementary Figure 1 is presented, but it really is an odd figure and does not add to the data at all. I think it should be removed. It is unusual to have a figure that just shows the proportion of sequences at each taxonomic level, without any additional details. There is also no legend for that figure so unless the reader has seen the text, the figure makes no sense.

From the tracked change document
line 628> misspelled Mississippi
line 755 maybe > may be
line 876, change "in contradiction" to "in contrast"
line 891, change "math" to "match"
line 894, I would change "severe" to something else like "critical" or "important"

---

## Round 0.2 · Minor Revisions

I agree with both reviewers that you did a good job in further improving your manuscript. Nevertheless, one of the reviewers provided a few remaining suggestions, particularly related to issues that need to be discussed in more detail.

Reviewer 1 ·

Basic reporting

N/A

Experimental design

Authors adequately addressed most of my questions. However, the are several new results that should be clarified before the acceptance of the manuscript; see REVIEWER RESPONSE 2 in the following sections:

REVIEWER:
Lines 350-355 and 385-387.
How many reads is the difference in the assignation to Vibrio species among using OTUs Vs ASVs?

AUTHORS:
59,645 reads mapped to Vibrio ASVs while 58,8443 reads mapped to Vibrio OTUs. Note that the % similarity used for read mapping in Vsearch is 97% so the difference in mapping statistics will likely be minimal despite the vast difference in the number of ASV vs OTU as read with mismatches will still map. We have included an excel file counts data for only Vibrio ASVs and OTUs in a new supplemental Table 5 which was used to construct Figure 6.

REVIEWER RESPONSE 2:
There is a great difference in the reads mapped to Vibrio OTUS 588,443, while only 59,645 reads mapped to ASVs. This enormous difference among methods should be clarified and discussed in the manuscript.


REVIEWER:
Please include the age and weight (g) of shrimps collected in methods, and discusses the potential effects of the shrimp age in defining the difference of the microbiota among Malaysian and Vietnamese shrimps (Figure 3).

AUTHORS:
We were able to request the age (but not weight as that was not measured during sampling) of shrimps collected for two farms and have included this information in Table 1 and have discussed briefly its effects on microbiota difference. Given that lack of detailed information from the farmers such as the shrimp diet, probiotic application etc, we are conservative in our discussion of the effects of shrimp age on microbiota difference.

REVIEWER RESPONSE 2:
Please discuss the potential effects of the size differences of shrimp (days post-hatching in Table 1) between the shrimp farms (97 days Vs 115 days Vs 65 days). It is well known that shrimp size/age have a strong impact on the microbiota structure, and I think that the clustering observed in Fig 3 is due to shrimp size and not by shrimp farm origin. What is the basis for choosing and comparing these three samples with a large size difference? This could potentially have a confounding effect on the results. Add some discussion points on this.


REVIEWER:
For a better comprehension of the shared and unique microbiomes between intestines and water from the same pond and between farms, the construction of Venn diagrams of shared ASVs at family and/or genus level will be very useful.

AUTHORS:
A Venn diagrams showing shared ad unique microbiomes at the family level has been constructed based on microbial families that are detected in > 90% of samples in any given group (Figure 4A). Additional information used to construct the Venn diagram is provided in Supplementary Data 5.

REVIEWER RESPONSE 2:
Please discuss the biological roles of families found shared and unique of the different Venn comparisons.

Validity of the findings

N/A

Additional comments

N/A

Reviewer 3 ·

Basic reporting

Well done; much better inclusion of references.

Experimental design

Well defined and rigorous.

Validity of the findings

Findings are well stated and presented.

Additional comments

The authors have addressed all my concerns and it would appear, the concerns of the other reviewers as well.

---

## Round 0.3 · accepted · Accept

Thanks for responding so efficiently and with great care to the concerns and suggestions of reviewers during the various stages of the manuscript. Nice piece of work.

# Reviewer 1 ·

Basic reporting

The authors performed the changes suggested. I strongly recommend it to be published in the journal.

Experimental design

The authors performed the changes suggested. I strongly recommend it to be published in the journal.

Validity of the findings

The authors performed the changes suggested. I strongly recommend it to be published in the journal.

Additional comments

The authors performed the changes suggested. I strongly recommend it to be published in the journal.